# Kinematic and Kinetic Characteristics of Repetitive Countermovement Jumps with Accentuated Eccentric Loading

**DOI:** 10.3390/sports10050074

**Published:** 2022-05-06

**Authors:** Micah Gross, Jan Seiler, Bastien Grédy, Fabian Lüthy

**Affiliations:** Department for Elite Sport, Swiss Federal Institute of Sport Magglingen (SFISM), Hauptstrasse 247, 2532 Magglingen, Switzerland; jan.seiler@baspo.admin.ch (J.S.); bastien.gredy@students.bfh.ch (B.G.); fabian.luethy@baspo.admin.ch (F.L.)

**Keywords:** reactive jump, repeated jumps, landing, stretch-shortening cycle (SSC), ground reaction force, electromyography (EMG)

## Abstract

Two methods for challenging the musculoskeletal and nervous systems to better exploit the stretch-shortening cycle (SSC) mechanism during plyometric training are reactive strength exercises and accentuated eccentric loading (AEL). Combining repetitive, reactive jumping with AEL poses a novel approach, in which the effects of both methods may be combined to elicit a unique stimulus. This study compared kinematic, kinetic, and electromyographic variables between a control (CON1) and two AEL conditions (AEL2 and AEL3). Additionally, non-reactive and reactive jumps performed within these sets were compared. Participants performed two sets of six countermovement jumps (CMJ) under each loading condition. AEL3 had moderate to large positive effects (es) on peak and mean eccentric force (es = 1.1, 0.8, respectively; both *p* < 0.01), and eccentric loading rate (es = 0.8, *p* < 0.01), but no effect on concentric variables or muscle activation intensity. The effects of AEL2 were similar but smaller. With or without AEL, there were moderate to large positive effects associated with reactive CMJ (second jump in a set, compared to the first) on peak and mean eccentric velocity (es = 1.7, 0.8, respectively; both *p* < 0.01), peak and mean eccentric force (es = 1.3, 1.2, *p* < 0.01), eccentric loading rate (es = 1.3, *p* < 0.01) and muscle activity (es = 1.8–1.9, *p* < 0.01). Concentric variables did not differ. Thus, the flight phase and act of landing during reactive jumps elicited greater increases in eccentric forces, loading rates, and muscle activity than AEL. Nonetheless, kinetic variables were greatest when AEL was combined with reactive jumping. Considering the limitations or complexity associated with most AEL protocols, sets of repetitive (reactive) CMJ may be more pragmatic for augmenting eccentric kinetic variables and neuromuscular stimuli during training.

## 1. Introduction

Quick and explosive movements play an important role in the performance of many sports. These movements include jumping, sprinting, and sharp changes of direction, and often involve stretch-shortening cycles (SSC) of the participating muscle-tendon systems. An SSC is a mechanism by which to enhance muscle power output by taking advantage of the elastic properties of tendons and muscle fibers. When agonist muscles for a particular movement are stretched (e.g., with a backswing or other countermovement) just prior to their contraction, muscle-tendon systems can act like springs, storing and re-releasing elastic energy—thus increasing movement economy, force production, and power output [1].

Effective utilization of recoil energy during SSC depends on several neural and anatomical factors. In any SSC, the nervous system attempts to integrate feedback (sensing of the stretch) and feedforward (intention of the movement) information to optimally balance muscle stiffness for maximal recovery of recoil energy, and muscle yielding to avoid tissue damage [2]. Nonetheless, this neural modulation must work within limits set by the overall strength (stretch-force tolerance) and contractile function (shortening force capacity) of a muscle-tendon system. Thus, training to improve SSC power output should aim to enhance eccentric strength and concentric contractile function, and therefore involve exposure to high eccentric loads or explosive concentric contractions. Accordingly, such training often includes plyometric exercises, which utilize rapid SSC movements to elicit both of these stimuli. A typical plyometric movement for the lower extremity is the vertical countermovement jump (CMJ).

Two methods for further challenging the musculoskeletal and nervous systems to better exploit the SSC mechanism during plyometric movements are reactive exercises and exercises with accentuated eccentric loading (AEL). Both of these methods increase eccentric loading beyond that of a mere countermovement, which may improve stretch-force tolerance. Moreover, these methods potentiate concentric force and power in some cases, which is believed to enhance the neuromuscular training stimulus [3]. It should be mentioned, however, that accentuating the eccentric load does not enhance concentric muscle output *per se* [4,5]. As mentioned above, stretch forces must be deemed manageable for tissues by the nervous system for this to occur; otherwise, muscle yielding is increased and the muscle-tendon system begins to act more like a shock absorber than a spring [1,2].

Reactive strength exercises such as depth jumps, drop jumps, or repetitive CMJ have been common practice for decades. These exercises comprise plyometric movements in which a flight phase precedes the SSC action. In this situation, the downward acceleration during flight and the act of landing increase the peak eccentric velocity, rate of loading, and muscle activation—and concentric output in some cases [3]—compared to plyometric contractions without a preceding flight phase. AEL has emerged more recently as a method for augmenting the mechanical and neuromuscular loads encountered during explosive and maximal strength exercises by using greater external resistance in the eccentric phase than in the concentric phase. Similar to reactive strength exercises, AEL has repeatedly been shown to enhance eccentric mean forces, eccentric peak velocity, mean and peak power, impulse, rate of force development, and muscle activation [6,7,8,9]—seemingly for a wide range of loads and movement velocities. Moreover, the detailed data presented in the study of Aboodarda et al. [10] suggest that with ballistic jumps, eccentric variables are augmented in relationship to the externally applied eccentric load. Concentric variables, such as peak power, force, velocity and jump height, benefit consistently from AEL in ballistic exercises with light or moderate (up to ~65% of maximum) concentric loads [10,11,12,13], but this does not appear to be the case for non-ballistic exercises [14], or where concentric resistance is high (beyond ~80% of maximum) and velocity relatively low [8,9,15,16]. 

Combining reactive strength exercises with AEL has not been investigated as thoroughly as each of these methods in isolation. The two available studies that reported kinetic or kinematic outcomes of reactive jumps (drop jumps) combined with AEL revealed clear increases in eccentric loading indicators, but no benefit to concentric muscle power or other variables. One of these studies used elastic bands and loads of ~20–30% of body weight, which enhanced the eccentric impulse and rate of force development (RFD) and increased quadricep activity across the eccentric phase, but had no effects on concentric kinetics, jump height, or muscle activation [6]. The other study employed drop jumps with AEL as a means to elicit post-activation potentiation. Although the authors provided no kinetic data, they mentioned that reactive jump height did not increase compared to the control condition [15]. Especially intriguing is the study of Aboodarda et al. [6], which combined two AEL loading conditions with three drop jump starting heights in a crossover design. Firstly, the results of that study suggests that somewhat similar enhancements to eccentric variables can be achieved by manipulating either the external eccentric load or starting height. Furthermore, the results indicate that the effects of AEL and starting height are additive.

These and most other available AEL studies have accentuated eccentric loads by means of either weights or elastic bands, which are released at the eccentric-to-concentric transition point. While these methods probably reflect most training settings the best, they also entail an inevitable delay of several seconds between repetitions in which weights or bands must be re-affixed manually. As these methods preclude uninterrupted, multi-repetition sets with AEL, most studies have employed either single repetitions with AEL [6,8,9,10,11,12,13,14,16], multi-repetition sets with AEL applied to the first repetition only, or cluster sets with multiple AEL repetitions separated by 30 s [15,17,18]. 

Nonetheless, performing continuous sets of any exercise, as opposed to isolated repetitions or cluster sets, elicits a unique stimulus—both biomechanically [19] and in terms of fatigue, e.g., [17,18]. Particularly for vertical jumps, Lam et al. [19] has shown some advantages of continuous sets in terms of joint power and extension velocity. Furthermore, although peak (non-SSC) landing forces were lower for consecutive jumps than for a single jump, eccentric forces within the actual SSC were most certainly higher following a flight phase than for an isolated jump [19]. 

Since eccentric loading within SSC is clearly enhanced by either AEL or by performing jumps consecutively, the question arises as to whether the effects of these two methods are additive, or whether combining them might otherwise elicit a unique stimulus. We are aware of only study that combined AEL with repetitive jumping. The results of that training intervention study suggested greater improvement in drop jump performance by adding AEL to jump squat training sets [20]. Furthermore, the authors noted that this training was especially effective for developing power in SSC movements commencing with higher eccentric forces (i.e., drop jumps more so than CMJ).

The study of Horwath et al. [20], being the only thus far to combine AEL with continuous jumping, revealed a unique work-around for the limitations of weight releasers or elastic bands. They employed an electronically controlled cable pull device that allows eccentric and concentric resistance to be programmed independently. While providing evidence for likely training effects of repetitive jumping with AEL, their study did not offer insight into the acute biomechanical differences between the training methods they investigated. Thus, the first aim of the current study was to compare eccentric and concentric velocity, eccentric force and loading rates, concentric force and power, and electromyographic variables between a control (non-AEL) condition and two AEL conditions during sets of repetitive CMJ. We hypothesized that these variables would be enhanced for all jumps in a given set, in direct relation to the eccentric load. This design also lent itself for a further comparison; therefore, the second aim was to compare the initial, non-reactive jump with the subsequent, reactive jump within a given set. Here, we hypothesized that the differences between non-reactive and reactive jumps would be at least as large as those between non-AEL and AEL jumps.

## 2. Materials and Methods

### 2.1. Participants

Ten basketball players, eight of which played professionally in the 1st Swiss Division, volunteered to participate in the study. They declared themselves to be uninjured at the time of the measurements. After having been informed about the aims and risks of the study, as well as the detailed procedures—which were approved by an institutional review board and conformed to the Declaration of Helsinki—players provided their written informed consent to participate. The mean (±standard deviation) age, body mass, and height of the participants were 25 ± 4 y, 88 ± 9 kg, and 1.93 ± 0.08 m, respectively.

### 2.2. Design

A repeated-measures (within-subject), randomized design was adopted to compare a control loading condition (CON1) with two AEL conditions (AEL2 and AEL3). Within one testing session, each participant performed two separate sets of CMJ under each of the three loading conditions. Participants performed these six sets in an individually randomized order. Differences between non-reactive CMJ and reactive CMJ (RCMJ) were also addressed in a repeated-measures manner by comparing the first and second jumps within the same set. Finally, the evolution of jump variables across the five RCMJ within each set were analyzed using regression analyses and compared in a repeated-measures manner.

### 2.3. Procedures

After arriving on site, participants completed a standardized warm up, consisting of 5–10 min of stationary cycling and mobility exercises. At the end of warm-up, they were instructed to perform a proper CMJ with hands placed on the hips (to inhibit arm contribution to the jump). They then performed two sets of 3–4 consecutive CMJ with no additional resistance. Subjects were encouraged to jump as high as possible and as fast as possible. After this warm-up, surface electrodes (F3010, FIAB, Vicchio, Italy) were placed on the thighs and lower legs (see details below). 

Between the warm-up and main measurements, subjects performed a total of six familiarization sets with AEL in the test setting. First, three sets of three consecutive CMJ with an additional concentric load of 7.5% body mass; whereas during the first set, no AEL was provided (eccentric load remained 7.5%), the second and third sets were performed with accentuated eccentric loads of 15 and 22.5% body mass, respectively. Familiarization ended with three further sets of three consecutive CMJ with an additional concentric load of 15% BM, and eccentric loads of 15%, 30% and 45% of body mass, respectively. These corresponded to the three experimental conditions. One minute of rest was afforded between familiarization sets. 

The main sets consisted of six jumps each. This set length was chosen for the study based on experience from the field, where the intention was to avoid substantial decreases in neuromuscular intensity. Participants were encouraged to perform each jump with maximal intensity. During the main measurements, three minutes of passive recovery were afforded between sets. A video clip of an example jump set is available in the Appendix A.

### 2.4. Loading and Measurement Instrumentation

Downward-directed loads were achieved with an electronically controlled, dual-sided cable pull training device (1080 Quantum Syncro, 1080 Motion, Stockholm, Sweden). This device uses two electric motors to apply tension to nylon ropes on the left and right sides in a synchronized manner. The device can track the linear motion (speed and direction) of the ropes continuously and modulate motor torque accordingly to maintain tensile force on the ropes at a programmed value. With the accompanying mobile application (1080 Motion for Windows, 1080 Motion, Stockholm, Sweden), the tensile force can be programmed for eccentric (rope reeling-in) and concentric (rope reeling-out) movement phases separately, although not entirely independently (eccentric force may not exceed three times the concentric force). 

Resistances were determined as fixed percentages of body mass. In order to achieve sufficient eccentric loads while working within the constrictions of the software application (maximal eccentric-concentric load ratio of 3:1), a constant concentric load of 15% (13.2 ± 1.3 kg) was combined with eccentric loads of 15% (CON1), 30% (AEL2), and 45% (AEL3) for the three loading conditions, respectively.

During familiarization and testing, participants wore a vest designed to attach to the Quantum’s ropes using carabiners. Ropes from the left and right sides were redirected at floor level by pulleys situated 0.8 m from the mid-line of the stance area and attached to the back of the vest at approximately mid-thorax height (depending on participant body height, between 1.04 and 1.25 m above the pulleys). As such, the ropes provided resistance in a diagonal direction, equally from sides. In addition to providing resistance, the Quantum recorded the one-dimensional rope speed and position (rope extension, in m) continuously at 333 Hz.

Ground reaction forces were recorded at 1000 Hz using a three-dimensional force plate (type 9260AA6, Kistler Instruments AG, Winterthur, Switzerland). Furthermore, the muscle activity of six leg muscles were recorded at 1000 Hz using a surface electromyography (EMG) device (M320, Myon AG, Switzerland). For EMG measurements, electrodes were placed on three muscles of each leg according to SENIAM recommendations (www.seniam.org, accessed on 1 November 2021). The muscles were the vastus lateralis (VL), vastus medialis (VM), and gastrocnemius medialis (GM). Data from the force plates and EMG were fed synchronously into the same software (Vicon Nexus, Version 2.9, Vicon Motion Systems Ltd., Oxford, UK).

### 2.5. Data Processing

Because the Quantum’s ropes were attached at an angle, the vertical movement component had to be calculated using the diagonal speed, the constant widths between the pulleys and vest attachment points, and basic trigonometry (see Figure 1). The vertical speed signal was then resampled to 1000 Hz and synchronized with the force plate and EMG data, which made it possible to divide the force signal into eccentric and concentric phases.

Synchronization for the eccentric phase was achieved by aligning the time point of the peak eccentric speed with the earliest time point at which net VGRF was positive. Likewise, synchronization for the concentric phase was achieved by aligning the time point of the peak concentric speed with the latest time point at which net VGRF was positive. Net VGRF was taken as the measured VGRF minus the phase-specific virtual weight. The phase-specific virtual weight for phases with a 15% load (concentric phase of all three conditions and eccentric phase of CON1) was equal to body weight plus combined downward rope force, which was determined from force plate data during the stance phase of a CON1 trial. For the eccentric phases of AEL2 and AEL3, this value was multiplied by 130/115 or 145/115, respectively. Mechanical power was calculated as the product of resultant force and vertical velocity.

The kinematic variables selected for addressing the research questions included the duration of eccentric and concentric phases (tecc, tcon, respectively). The upward acceleration phase was delimited by the time points of the peak eccentric and peak concentric speed; this essentially corresponded to the ground contact phase for jumps 2–6. Furthermore, the mean and peak speed of the eccentric (vecc_mean_, vecc_peak_) and concentric phases (vcon_mean_, vcon_peak_) were determined. The kinetic variables were the mean and peak eccentric phase force (Fecc_mean_, Fecc_peak_), as well as the mean concentric phase force Fcon_mean_ and the mean and peak concentric phase power (Pcon_mean_, Pcon_peak_). Finally, the time (tFecc_peak_) and average rate of force development (RFDecc) between the onset of upward acceleration and the first peak in eccentric force were calculated. For the first non-reactive CMJ, the peak force was near the end of the eccentric phase. However, the eccentric phases of RCMJ typically displayed two peaks; therefore, for the determination of the Fecc_peak_, tFecc_peak_, and RFDecc of RCMJ, the first peak in force—which corresponded to the peak eccentric power—was used. Each of these parameters was generated for each jump of each set.

After having subtracted the raw signal mean from all data points, EMG signals were band-pass filtered (4th order Butterworth) with a cut-off frequency range of 20–450 Hz. Thereafter, the signals were rectified and filtered again with a low-pass Butterworth filter (4th order) and a cut-off frequency of 10 Hz. Finally, the signals were normalized to the peak value of the entire measurement session for the given participant. From the normalized signals, the mean of the entire upward acceleration phase (see above) of each jump was calculated.

### 2.6. Statistics

Statistical operations were performed using custom Python scripts based on the scipy, statsmodels, and scikit_posthocs libraries. For all significance testing, alpha was set to 0.05. 

The main effects of the loading conditions were assessed using one-way repeated-measures tests (ANOVA if normal distribution was deemed probable by the Shapiro–Wilk test; otherwise, Friedman tests). These analyses were made using the five RCMJ (jumps 2–6) for each set; thus, *n* was 100 jumps (10 participants with two sets each and five analyzed jumps per set) for each loading condition. Significant differences between loading conditions were identified with Bonferroni (with ANOVA) or Nemenyi (with Friedman test) post-hoc analyses. Standardized differences (effect sizes, es), which we report only in the case of a significant effect, were calculated as the differences of means normalized to the pooled standard deviation, and classified according to Hopkins et al. [21] as trivial (0–0.19), small (0.2–0.59), moderate (0.6–1.19), large (1.2–1.9) very large (2–3.9), or extremely large (≥4). Post-hoc power analyses for these comparisons, performed with G*Power software (Version 3.1) [22], indicated that 80% statistical power was achieved for effect sizes of 0.8 and above. The statistical power for effect sizes of 0.6 was 55%.

Differences between the CMJ and the first RCMJ were assessed with repeated-measures comparisons (*t*-test if normal distribution was deemed probable by the Shapiro–Wilk test; otherwise, a Wilcoxon test). By pooling all six sets (i.e., all three loading conditions) from all 10 participants, the *n* was 60 for each of these comparisons. The effect sizes for each pair of loading conditions were calculated and interpreted as described above. Post-hoc power analyses for these comparisons indicated that 80% statistical power was achieved for effect sizes of 0.7 and above. The statistical power for effect sizes of 0.6 was 65%.

## 3. Results

Descriptive data from the three loading conditions (pooled data for RCMJ, i.e., jumps 2–6) are displayed in Table 1 as mean ± sd. Descriptive data for jumps 1, 2, and 6 of a set (pooled from all three loading conditions) are displayed in Table 2. 

### 3.1. AEL and Loading Magnitude (Pooled RCMJ)

Significant effects of the loading condition were identified for eccentric force and loading rate variables (CON1 < AEL2 < AEL3). Post-hoc analyses and effect sizes indicated only small differences in Fecc_mean_ (es = 0.39, 0.46) and RFDecc (es = 0.40, 0.42), but moderate differences in Fecc_peak_ (es = 0.61, 0.73) between CON1 and AEL2 and between AEL2 and AEL3. Differences in Fecc_mean_, Fecc_peak_, and RFDecc between AEL3 and CON1 were moderate (es: 0.77–1.14). Increases in RFDecc with the loading condition were due not only to greater Fecc_peak_—there were also small effects on tFecc_peak_. 

Additional significant effects of the loading condition were observed for vecc_mean_ (post-hoc analyses revealed: AEL3 < CON1 and AEL2] and vecc_peak_ (AEL2 and AEL3 > CON1), although differences between conditions, if significant, were small. No significant effects of loading conditions existed for concentric kinematic or kinetic variables. 

Representative force-time curves for each loading condition, highlighting differences in Fecc_peak_ and RFDecc, are displayed in Figure 2. 

There was a significant effect on muscle activity by VM, whereby post-hoc tests revealed greater activity for AEL3 than for CON1. However, differences between AEL3 and CON1 were small (es = 0.29). No effects of loading condition were observed on the activity of VL or G.

### 3.2. CMJ vs. RCMJ (Pooled Data from All Loading Conditions)

The kinematic comparison of the first (CMJ) and second (RCMJ) jumps within sets revealed a large difference in vecc_peak_ (es = 1.74, *p* < 0.001) and a moderate difference in vecc_mean_ (es = 0.76, *p* < 0.001). In terms of kinetics, large differences (es: 1.2–1.3) were apparent for eccentric forces and loading rates. Indeed, RCMJ displayed greater Fecc_mean_ (es = 1.2, *p* < 0.001) and RFDecc (es = 1.3, *p* < 0.001). The increase in RFDecc was due to the attainment of greater Fecc_peak_ (es = 1.3, *p* < 0.001) in less time (effect size on tFecc_peak_ = 1.8, *p* < 0.001).

On the other hand, tcon, vcon_mean_, Fcon_mean_, and Pcon_mean_ did not differ between CMJ and RCMJ. Moreover, vcon_peak_ and Pcon_peak_ were significantly lower (*p* ≤ 0.001) for the RCMJ, although differences were small (es = 0.35, 0.42, respectively). Representative force-time curves for the first, second and last jumps from a set, highlighting differences in Fecc_peak_ and RFDecc, are displayed in Figure 3.

Differences in the EMG activity of all three muscles were large (for each, es = 1.9), with muscle activation being significantly greater (*p* < 0.001) for the RCMJ. 

## 4. Discussion

### 4.1. AEL and Loading Magnitude

In accordance with our hypothesis, increasing the external downward force during the downward phase of repeated jumping led to a greater velocity at the onset of the braking phase. As the braking distance remained essentially the same regardless of the loading condition, greater peak and mean forces—as well as rates of force development—were attained in the eccentric phase by accentuating eccentric loading. These variables increased proportionally to the increase in external eccentric load in AEL2 and AEL3. This is in line with the majority of previous studies investigating AEL and lower body exercises [6,7,9,17]. 

Additional eccentric load in the two AEL conditions was double or triple that of the control condition (where it was 15% body weight). This represented an increase of 13 and 26% in total load for AEL2 and AEL3, respectively. These were close to the 12% and 31% differences in Fecc_peak_ observed for these two conditions, respectively, which suggests a close relationship between total load and peak landing force. Differences in eccentric mean force were, however, smaller (4 and 9%, respectively). Although RFD_ecc_ increased by a larger degree on average (by 34 and 81%), the variation among participants was substantial. Thus, the overall effects on RFD_ecc_ must be interpreted as small for AEL2 and moderate for AEL3. The changes in EMG activity that occurred with increasing eccentric loads were also small.

High mechanical loads are in general recognized to stimulate muscle hypertrophy [23] as well as positive adaptations to passive tissue [24,25]. The performance-related benefits of high eccentric loads specifically, such as during the braking phase of repetitive squat jumps, were demonstrated in a novel study by Hori et al. These authors suggested that eccentric loading during jump training plays an important role for improving concentric force at high movement velocities, whereas this is not the case for improvements at slower velocities [26]. Thus, based on the observation that the investigated AEL protocols predominantly increase eccentric variables, this method may be best suited for stimulating structural adaptations or high-velocity concentric force. 

In the current study, there was no evidence that increasing the eccentric load helped enhance concentric muscle output. This was in contrast to some previous studies [9,10,11,27]. One factor that may have played a role in the present study, in contrast to previous research, is that our jumps were performed with an additional concentric load, which was unavoidable with the setup we employed. Another issue to consider is that, despite performing three familiarization sets prior to data acquisition, participants had not been otherwise exposed to AEL in previous sessions. Nonetheless, there was also no apparent detriment to concentric neuromuscular performance (force, velocity, or power) associated with greater eccentric loading in the studied athletes. 

During eccentric muscle work, including SSC, muscle-tendon systems function somewhere along the continuum between springs and shock absorbers [1] depending on the desired outcome and also on the external environmental factors. As mentioned in the introduction, augmented eccentric loading—whether by AEL or, for example, by increasing the drop height of drop jumps—does not increase concentric force or power per se. The potentiation of concentric SSC output is dependent on strength boundaries and the neuromuscular system’s ability to modulate stiffness within these boundaries so as to better store and release recoil energy (spring function), while protecting tissues (shock absorber function) from overload damage [2]. Apparently, the strength and/or the experience level of the current cohort did not facilitate the potentiation of concentric output. To which degree this might change over time or differ for other athlete populations requires further investigation.

### 4.2. CMJ vs. RCMJ

Based on the comparison between the first, non-reactive CMJ and the second RCMJ within the same set, it is clear that performing reactive countermovement jumps provides a greater mechanical and neuromuscular stimulus during the eccentric phase than single countermovement jumps from a standing start. For example, reactive jump execution augmented Fecc_peak_ by 40% in CON1 and by 79% in AEL3. In comparison, AEL3 increased Fecc_peak_ by only 9% in CMJ and 40% in RCMJ (Figure 4). This confirms our hypothesis that reactive jumps are at least as effective as AEL for enhancing eccentric mechanical stimuli. This finding is supported by the study of Aboodarda et al. [6], the results of which suggest that manipulating drop height during drop jumps achieves similar or greater enhancements to eccentric kinetic variables (force, loading rate) as AEL. 

On the other hand, the higher forces and loading rates in the stretch phase did not enhance, but rather inhibited muscular output in the shortening phase, as seen in reduced concentric peak velocity and power. Although AEL did not enhance concentric output in the current study either, our hypothesis of equal effectivity for reactive jumps and AEL did not hold true. Thus, reactive countermovement jumps may enhance eccentric loading and improve braking ability more effectively than countermovement jumps starting on the ground, with or without AEL. However, reactive jumps were less effective for maintaining or enhancing concentric variables or otherwise facilitating greater SSC performance in the studied athlete cohort.

### 4.3. Limitations

The main limitations of the current study were that jumps were performed with an additional concentric load, and that participants (despite familiarization in the measurement session) were not otherwise accustomed to AEL. Based on previous research and observations from the field, both of these may have precluded positive effects of one or both AEL protocols on concentric muscle output. A further limitation is that the investigated protocol was only possible on electronically controlled devices that allowed concentric and eccentric loads to be programmed separately. However, technology like the device we employed is becoming more common in strength and conditioning settings—particularly for competitive athletes.

## 5. Conclusions

In conclusion, the investigated AEL protocols successfully augmented braking forces in relatively unaccustomed athletes. For the two AEL conditions, the eccentric forces and rates of force development increased in relation to additional eccentric load. No acute affects, either positive or negative, were apparent for concentric variables. This study also shows that the reactive execution of jumps increased eccentric forces, loading rates and muscle activity more than AEL. Nonetheless, the effects on the eccentric kinetics of reactive execution and AEL were additive (Figure 2, Figure 3 and Figure 4). Thus, in light of the complexity associated with some AEL protocols, sets of repetitive (reactive) CMJ may be more pragmatic than AEL for augmenting eccentric kinetic variables and neuromuscular stimuli during training.

## Figures and Tables

**Figure 1 sports-10-00074-f001:**
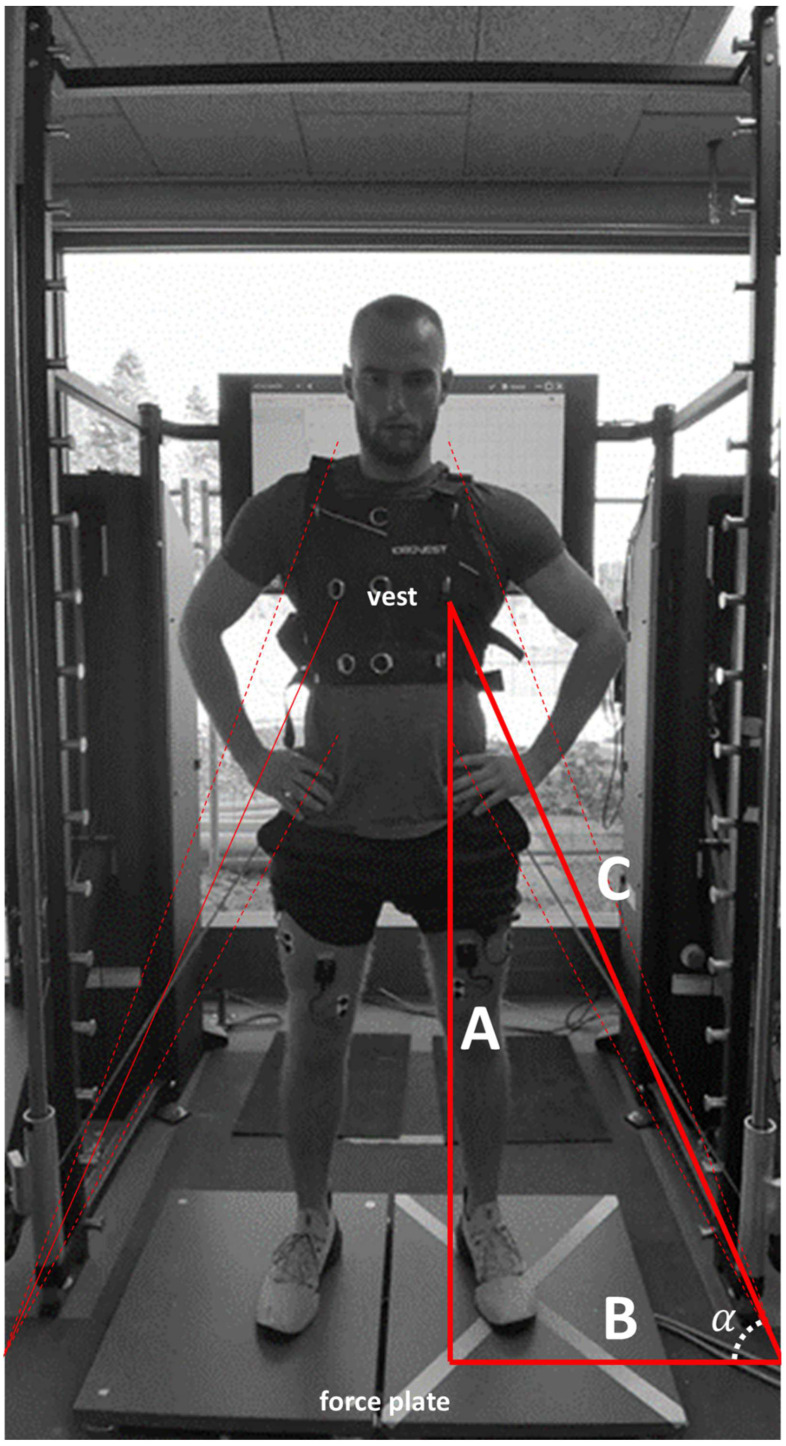
Representation of the test setup. Resistance while jumping was provided by the ropes of the 1080 Quantum Syncro device, which also measured rope extension and retraction speed. Participants wore a vest, to which ropes from the left and right sides were attached to the mid-back with carabiners. To obtain vertical movement velocity, a right triangle with sides A, B, and C was assumed to have a constant width (B) delimited by the floor-level pulley and the vest attachment point. Using the constant width (B) and the variable rope extension length (C) at any given time point, the angle between the floor and the rope (${cos}^{-1}\frac{B}{C}$) was calculated. Thereafter, the vertical velocity was calculated as $\sin\alpha \cdot s_{d}$ where $s_{d}$ was the diagonal rope speed (i.e., $\frac{∆C}{∆t}$ ) obtained from the 1080 Quantum. Dashed lines indicate the ropes’ positions (and thus varying extension length) at the bottom and top of a jump.

**Figure 2 sports-10-00074-f002:**
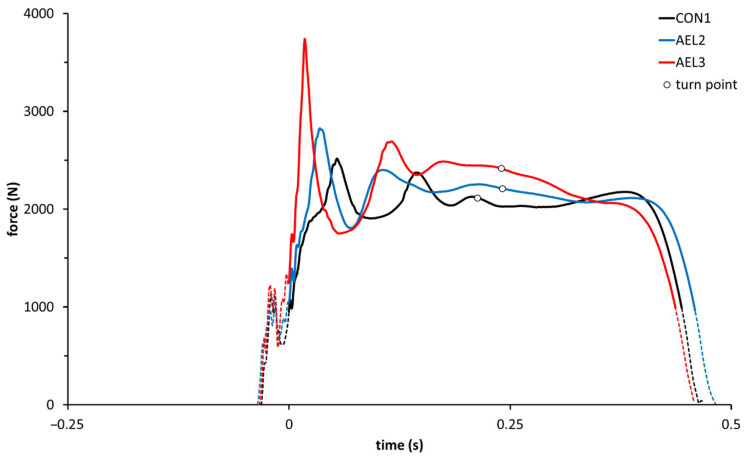
Representative force-time curves for a reactive countermovement jump (RCMJ) under each of the three loading conditions. CON1, AEL2, and AEL3 designate jumps with concentric loads of 15% of body mass, but eccentric loads of 15%, 30%, and 45% of body mass, respectively. The bold portion of each line (beginning at time = 0) represents the positive acceleration phase, i.e., that where the ground reaction force exceeded body weight. The turn point (○) represents the transition from the braking (eccentric) phase to the propulsive (concentric) phase. Particularly noteworthy are the increases in eccentric peak force and decrease in time to the attainment thereof with increasing eccentric load. Further, there are no (significant) differences in concentric force.

**Figure 3 sports-10-00074-f003:**
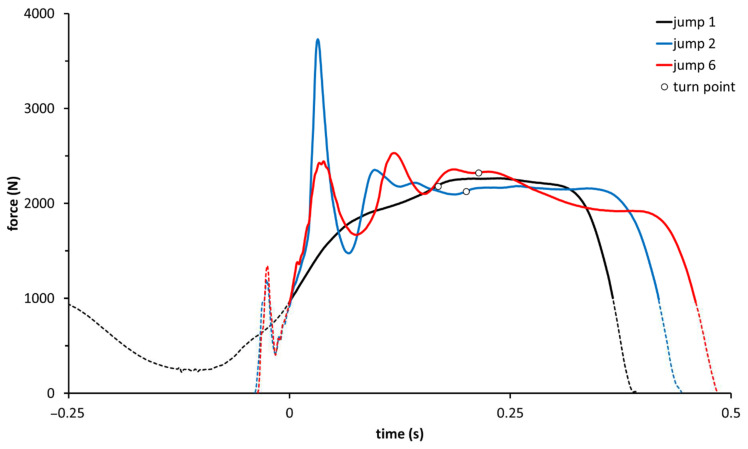
Representative force-time curves for the first (countermovement jump from a standstill, CMJ), as well as the second and last (reactive countermovement jumps, RCMJ) jumps from a six-jump set. The bold portion of each line (beginning at time = 0) represents the positive acceleration phase, i.e., that where the ground reaction force exceeded body weight. The turn point (○) represents the transition from the braking (eccentric) phase to the propulsive (concentric) phase. Of note are the generally greater eccentric forces and particularly the earlier and greater force peak for jumps 2 and 6 (RCMJ) compared to jump 1 (CMJ). There is also a slight but clear decline in eccentric peak force from jump 2 to jump 6. Concentric forces do not differ between jumps 1 and 2, but are slightly lower for jump 6.

**Figure 4 sports-10-00074-f004:**
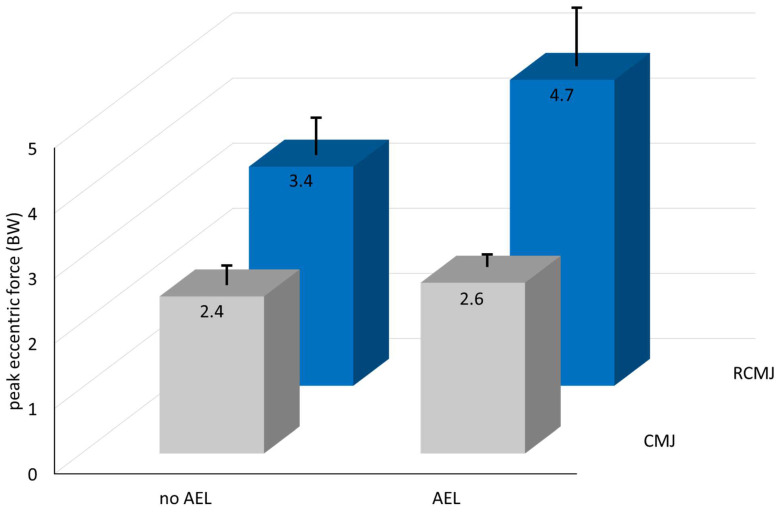
Differential effects of accentuated eccentric loading (AEL) and of countermovement jumps (CMJ) executed in a reactive manner (RCMJ) on peak eccentric force. The labels ‘no AEL’ and ‘AEL’ represent the CON1 and AEL3 conditions described in the text. The effects of AEL are shown from left to right for both CMJ and RCMJ. The effects of reactive execution are shown from front to back. The data suggest that reactive execution affects peak eccentric force more than AEL, but that the effects are additive when the two methods are combined. This was the case for other eccentric kinetic variables as well.

**Table 1 sports-10-00074-t001:** Descriptive data for the three loading conditions.

Phase	Variable	CON1	AEL2	AEL3
eccentric	Fecc_mean_ (BW) ^t^	2.3 ± 0.2	2.4 ± 0.2 *	2.5 ± 0.2 **
	Fecc_peak_ (BW) ^W^	3.0 ± 0.4	3.4 ± 0.5 **	4.1 ± 0.7 **
	tFecc_peak_ (s) ^t^	0.04 ± 0.01	0.04 ± 0.01	0.03 ± 0.01
	RFDecc (BW/s) ^W^	56 ± 30	74 ± 47 *	101 ± 57 **
	vecc_mean_ (m/s) ^t^	0.8 ± 0.1	0.8 ± 0.1	0.8 ± 0.1
	vecc_peak_ (m/s) ^t^	1.6 ± 0.1	1.7 ± 0.1 *	1.7 ± 0.1 *
	tecc (s) ^W^	0.23 ± 0.05	0.25 ± 0.04 *	0.26 ± 0.05 *
concentric	Fcon_mean_ (BW) ^t^	2.3 ± 0.1	2.3 ± 0.1	2.3 ± 0.2
	Pcon_mean_ (W/kg) ^t^	2.0 ± 0.2	2.0 ± 0.2	2.0 ± 0.2
	Pcon_peak_ (W/kg) ^t^	3.4 ± 0.3	3.4 ± 0.2	3.3 ± 0.3
	vcon_mean_ (m/s) ^W^	0.9 ± 0.1	0.9 ± 0.1	0.9 ± 0.1
	vcon_peak_ (m/s) ^W^	1.7 ± 0.1	1.7 ± 0.1	1.7 ± 0.1
	tcon (s) ^t^	0.21 ± 0.03	0.22 ± 0.03	0.22 ± 0.03
entire ground contact	G (%) ^t^	80 ± 6	78 ± 4	78 ± 7
	VL (%) ^t^	80 ± 5	81 ± 5	82 ± 5
	VM (%) ^W^	78 ± 6	79 ± 8	81 ± 10

Data are mean ± standard deviation, pooled for all reactive countermovement jumps (jumps 2–6 of a set) performed under a given loading condition. CON1, AEL2, AEL3 designate three loading conditions (details in text). For each condition, *n* = 100 jumps (10 participants, 2 sets each, 5 jumps per set). Fecc, vecc, tecc: force, velocity, and duration during the eccentric phase of jumps, respectively. tFeccpeak, RFDecc: time to peak eccentric force and the rate of force development from the onset of the eccentric phase up to peak eccentric force, respectively. Fcon, vcon, tcon: force, velocity, and duration during the concentric phase of jumps, respectively. BW: force variables are expressed as factors of body weight. G, VL, VM: mean muscle activation during the ground contact phase of the muscles gastrocnemius, vastus lateralis, and vastus medialis, respectively, expressed as a percentage of the individual session maximum for that muscle. ** indicates a significant difference and moderate effect size compared to CON1. * indicates a significant difference and small effect size compared to CON1. Superscripts to the right of variable names indicate whether comparisons were performed with repeated-measures *t*-tests (^t^) or the Wilcoxon test (^W^).

**Table 2 sports-10-00074-t002:** Descriptive data for selected repetitions of the six-jump sets.

Phase	Variable	Jump 1 (CMJ)	Jump 2 (RCMJ)	Jump 6 (RCMJ)
eccentric	Fecc_mean_ (BW) ^t^	2.0 ± 0.2	2.4 ± 0.3 **	2.3 ± 0.2
	Fecc_peak_ (BW) ^W^	2.5 ± 0.3	3.9 ± 1.1 ***	3.3 ± 0.9
	tFecc_peak_ (s) ^W^	0.20 ± 0.06	0.03 ± 0.01	0.04 ± 0.02
	RFDecc (BW/s) ^W^	6.7 ± 2.8	95 ± 72 ***	73 ± 55
	vecc_mean_ (m/s) ^t^	0.7 ± 0.1	0.8 ± 0.1 **	0.8 ± 0.1
	vecc_peak_ (m/s) ^W^	1.1 ± 0.2	1.6 ± 0.1 ***	1.7 ± 0.1 *
	tecc (s) ^W^	0.22 ± 0.07	0.26 ± 0.06 **	0.22 ± 0.07
concentric	Fcon_mean_ (BW) ^t^	2.4 ± 0.2	2.4 ± 0.2	2.2 ± 0.1
	Pcon_mean_ (W/kg) ^t^	2.1 ± 0.3	2.0 ± 0.3	1.9 ± 0.3
	Pcon_peak_ (W/kg) ^W^	3.5 ± 0.3	3.4 ± 0.3	3.2 ± 0.3
	vcon_mean_ (m/s) ^W^	0.9 ± 0.1	0.9 ± 0.1	0.9 ± 0.1
	vcon_peak_ (m/s) ^W^	1.7 ± 0.1	1.7 ± 0.1	1.7 ± 0.2
	tcon (s) ^t^	0.21 ± 0.03	0.21 ± 0.04	0.24 ± 0.03 **
entire ground contact	G (%) ^W^	23 ± 4	76 ± 11 ***	80 ± 11
	VL (%) ^W^	24 ± 5	80 ± 8 ***	82 ± 8
	VM (%) ^W^	25 ± 7	76 ± 11 ***	81 ± 12 *

Data are mean ± standard deviation, pooled for all loading conditions. CMJ: countermovement jump performed from a standstill. RCMJ: reactive countermovement jump, preceded by a flight phase. For each variable *n* = 60 jumps (10 participants, 3 loading conditions, 2 sets per loading condition). Fecc, vecc, tecc: force, velocity, and duration during the eccentric phase of jumps, respectively. tFecc_peak_, RFDecc: time to peak eccentric force and the rate of force development from the onset of the eccentric phase up to peak eccentric force, respectively. Fcon, vcon, tcon: force, velocity, and duration during the concentric phase of jumps, respectively. BW: force variables are expressed as factors of body weight. G, VL, VM: mean muscle activation during the ground contact phase of the muscles gastrocnemius, vastus lateralis, and vastus medialis, respectively, expressed as a percentage of the individual session maximum for that muscle. With respect to column immediately to the left, *** indicates a significant difference and large effect size, ** indicates a significant difference and moderate effect size, and * indicates a significant difference and small effect size. Superscripts to the right of variable names indicate whether comparisons were performed with repeated-measures *t*-tests (^t^) or the Wilcoxon test (^W^).

## Data Availability

Not applicable.

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
