# Peer review of "Kinematic and Kinetic Characteristics of Repetitive Countermovement Jumps with Accentuated Eccentric Loading"

_sports, 2022, doi:10.3390/sports10050074_

Round 1
Reviewer 1 Report
General comments
The authors aimed to compare kinematic, kinetic, and EMG variables between two accentuated eccentric loading conditions and control condition. Additionally, non-reactive and reactive jumps with all sets were compared. The authors present a very interesting paper with a complete and justified introduction.
However, there are several methodological aspects that must be clarified to make sense of the study. Parts of contents in the manuscript are wordy and hard to read. Moderate editing of the manuscript and concise writing is required.
Specific Comments
Abstract
Page 1, line 2
The title is suggested to revise to “Kinematic and kinetic characteristics of repetitive countermovement jumps with accentuated eccentric loading”
Page 1, line 16, Line 18-19
ES and p value should be given in abstract.
Page 1, line 22
I am not sure what you mean by ‘the effects of landing and AEL were additive’.
Page 1, line 26
I suggested countermovement jumps (CMJ) as one of the key words instead of “consecutive jumps”.
Introduction
Page 2, line 86-89
You have not made it clear to describe reactive jump height was either reduced or unaffected by three AEL conditions compared to the control conditions….
Page 3, line 103
“consecutive” should be “consecutive jumps”. “single jumps” should be “a single jump”.
Page 3, line 108-111.
This is a very long sentence that is difficult to follow. I think you could split this sentence. (Page 2, line 53-56;…. Page 12, line 354-358, page 360-363………;page 13, line 400-404 as well).
Page 3, line 126
I suggested the authors to state one or two hypothesis.
Materials and Methods
Page 3, line 130-131
They were uninjured at the time of the measurements according to what criteria or any clinic professional?
Page 3, line 132-134
Was this study approved by an ethics committee?
Page 4, line 155-157
Are familiarization set composed of six sets of three consecutive CMJs? Please clarify.
Page 4, line 163-165.
Is each main testing set composed of six jumps? And each participant performed many testing sets as many repetitions as possible? If so, which one set did the author choose for the data analysis? Or just one main set to perform? It is suggested that the authors make a figure or diagram to demonstrate and clarify.
Page 4, line196
Please add abbreviations for vastus lateralis, vastus medialis, and. gastrocnemius medialis. In addition, ‘m’ should be deleted.
Page 7, line 231
Fmeancon should be “Fconmean”.
Page 7, line 264-266.
Is each variable for jumps 2-6 of ten subjects suitable for least-square linear regression? Please clarify.
Results
Page 8, Table 1
Fmeancon should be “Fconmean”.
All abbreviation should be listed in full name in the caption of the table as well.
Deleted “mean muscle activation during the ground contact phase is expressed as a percentage of the individual session maximum”.
”
Page 9 Table 2
Fmeancon should be “Fconmean”.
Deleted “mean muscle activation during the ground contact phase is expressed as a percentage of the individual session maximum”.
Page 9, line 280
Please describe which variables have a normal distribution and others haven’t.
Page 9, line 15.
Differences in these variables between….. What did the variables involve? Please describe.
Page 10, Figure 2
Use different colors to mark force-time curves for RCMJ under three loading conditions for better recognition.
Page 10, line 308-309
The authors described that “ In terms of kinetics, large differences (es: 1.2 – 1.3) were apparent for eccentric forces and loading rates. Indeed, RCMJ displayed greater Feccmean (es = 1.2, p<0.001), as well as attainment of greater Feccpeak (effect size = 1.3, p<0.001) in less time (effect size on tFeccpeak = 1.8, p<0.001)”. However, I didn’t see the analysis result of loading rates in this paragraph.
Page 11, line 318
I suggest authors to draw out a least-square linear regression figure for at least one variable and mark the linear regression equation on the figure.
Page 11, figure 3.
Use different colors to mark force-time curves for RCMJ under three loading conditions for better recognition.
Discussion
Page 13, line 403
The authors described that “…. but appear to have been ineffective at eliciting greater shortening velocities or otherwise facilitating…. ”
I am not sure what the “greater shortening velocities” means. Are these greater concentric velocities?
Page 13, line 420
The authors proposed that “either of these (reduced elastic stretching or fatigue) could explain the slight decrease in concentric output during SSC….” I suggest the authors analyze EMG amplitude and median frequency for VM, VL, and Gastrocnemius to detect if muscle fatigue or not.
Conclusions
Page 13, line 433
This paragraph is too long and wordy. Please rewrite concisely.
Author Response
Please see our point-by-point response to your review in the attached Word file.

Reviewer 2 Report
GENERAL COMMENTS.
In general, the manuscript purpose is interesting, i.e., to compare kinematic, kinetic, and electromyographic variables between a control (CON1) and two AEL conditions (AEL2 and AEL3). Additionally, non-reactive and reactive jumps with the sets were compared. However, I have some questions regarding statistical analysis. First, the size of sample is very restrict to confirm this results, and secondly, I suggest to the authors to change the resolution of figures of at least 300 dpi.
SPECIFIC COMMENTS
Introduction. Specify the rational of the study and the hypothesis has to be clarified.
Methods
Participants: The major weakness of this study was the sample size. So please add the results of effect size and power simulation of this study (using G*power for example) to prove that this sample was sufficient to have 80% of statistical power.
Statistical analyses
- Please add ICC and magnitude
Results.
- I suggest to the authors to add 2 columns at figure 1 and 2 and include “F” for ANOVAs and p values.
Discussion.
- The discussion can be tightened up. As is, it lacks focus in many areas for example the specific purpose and findings of this study
- More notes about practical application associated to the results of the study could be implemented.
Conclusion.
The conclusion to be reformulated (please try to make it shorter).
Author Response

(The authors gave the same response as above.)
